# Application of *Campylobacter jejuni* Phages: Challenges and Perspectives

**DOI:** 10.3390/ani10020279

**Published:** 2020-02-11

**Authors:** Leonid Ushanov, Besarion Lasareishvili, Irakli Janashia, Andreas E. Zautner

**Affiliations:** 1Institute of Veterinary Medicine, Agricultural University of Georgia, 0159 Tbilisi, Georgia; l.ushanoff@agruni.edu.ge (L.U.); b.lasareishvili@agruni.edu.ge (B.L.); 2Institute of Entomology, Agricultural University of Georgia, 0159 Tbilisi, Georgia; i.janashia@agruni.edu.ge; 3Institute of Medical Microbiology, University Medical Center Göttingen, 37075 Göttingen, Germany

**Keywords:** *C. jejuni*, bacteriophage, phage-based applications, phage therapy, *Campylobacter* phages

## Abstract

**Simple Summary:**

*Campylobacter jejuni* is a zoonotic bacterial pathogen found in abundance, e.g., in raw poultry. *C*. *jejuni* is part of the avian gut microbiome, causing no harm to birds. When birds are slaughtered, the bacterium is released from the intestines and contaminates the meat. Cross-contaminated foods processed in parallel to the meat cause diarrhea in humans and in rare cases may cause post-infectious complications like arthritis and paralysis of peripheral nerves. Illnesses caused by *C*. *jejuni* have been on the rise in both developed and developing countries. At the same time, *C*. *jejuni* has acquired resistance to important antibiotics, which further complicates therapeutic approaches. To decrease the *Campylobacter* load on chicken carcasses, chemical or thermal treatment can be used. However, this may change the taste or affect the appearance of chicken meat. Alternative treatments include application of probiotics active against *C*. *jejuni* to live chickens or the use of bacteriophages (viruses killing bacteria) to decrease *Campylobacter* in the guts of live chickens prior to killing. This article discusses phage-based applications already in use against other bacterial pathogens and specifically highlights the challenges associated with the use of *Campylobacter* phages, emphasizing the ways to overcome these challenges based on the existing research.

**Abstract:**

Bacteriophages (phages) are the most abundant and diverse biological entities in the biosphere. Due to the rise of multi-drug resistant bacterial strains during the past decade, phages are currently experiencing a renewed interest. Bacteriophages and their derivatives are being actively researched for their potential in the medical and biotechnology fields. Phage applications targeting pathogenic food-borne bacteria are currently being utilized for decontamination and therapy of live farm animals and as a biocontrol measure at the post-harvest level. For this indication, the United States Food and Drug Administration (FDA) has approved several phage products targeting *Listeria* sp., *Salmonella* sp. and *Escherichia*
*coli.* Phage-based applications against *Campylobacter*
*jejuni* could potentially be used in ways similar to those against *Salmonella* sp. and *Listeria* sp.; however, only very few *Campylobacter* phage products have been approved anywhere to date. The research on *Campylobacter* phages conducted thus far indicates that highly diverse subpopulations of *C. jejuni* as well as phage isolation and enrichment procedures influence the specificity and efficacy of *Campylobacter* phages. This review paper emphasizes conclusions from previous findings instrumental in facilitating isolation of *Campylobacter* phages and improving specificity and efficacy of the isolates.

## 1. Significance of *Campylobacter jejuni* as a Pathogen

*Campylobacter jejuni* is a causative agent of bacterial enteritis in humans with serious consequences for children, the elderly and the immunocompromised [1,2,3]. One of the rare sequelae associated with *C. jejuni* is Guillain–Barré syndrome (GBS), manifested as demyelination of the peripheral nerves due to autoimmune reaction and paralysis [4,5]. About 34%–49% (≈42%) of GBS cases are likely to be associated with prior *Campylobacter* infection [6]. Further post-campylobacteriosis sequelae are reactive arthritis, *Erythema nodosum*, and inflammatory bowel disease [6,7].

The avian reservoir is the predominant source of *C. jejuni* infections, which have been strongly linked to contaminated retail chicken by numerous studies [8,9]. As a commensal organism with high prevalence (10^9^ CFU/g) in chicken ceca, *C. jejuni* easily contaminates carcasses of slaughtered birds [10]. Davis and colleagues demonstrated experimentally that *Campylobacter* survives well on both chicken skin and meat at refrigerated temperatures [11].

Cases of *C. jejuni*-associated food poisoning have been rising worldwide. Studies from Brazil, Great Britain, Canada, Italy, Korea, Australia and Japan report contamination rates ranging from 20% to 80% [12,13,14,15]. According to the European Food Safety Authority (EFSA), *Campylobacter* continues to be the most commonly reported human gastrointestinal bacterial pathogen in the European Union (EU) since 2005 [16]. *C. jejuni* infections occurring each year cause close to 350,000 cases of disability and cost 2.4 billion Euros yearly [17]. There were 246,571 confirmed cases of Campylobacteriosis reported in the EU in 2018 [18].

Currently, there is no single effective intervention method for reduction of bird colonization by *C. jejuni*. Biosecurity measures only achieve partial clearance [19]. This was effectively demonstrated by a study conducted in central Italy where *Campylobacter* isolates were obtained, both by cloacal swabs and at the post-harvest stage, from farm chicken flocks kept under good sanitary practices [20]. Colonization of broiler chicks by *C. jejuni* takes place within the first three weeks after hatching, and, since *Campylobacter* colonization is not associated with signs of disease in chickens, horizontal spread of the pathogen usually remains unnoticed [9,21]. Combination of strict biosecurity, good manufacturing practice (GMP), hazard analysis and critical control points (HACCP) attempts can alleviate contamination levels but do not lead to a complete elimination of the bacteria. It is difficult to quantify the effectiveness of individual measures because they are dependent on very many interrelated local factors [17]. Additionally, there is no vaccination available against *C. jejuni* due to serological diversity of the pathogen and the short lifespan of broilers [19]. The estimated public benefits in terms of a higher reduction of the disease burden of campylobacteriosis and its sequelae are greater if efforts are made towards controlling *C. jejuni* during the primary production stage [17].

To control *C. jejuni*, scientists have proposed several methods based on either supplementing chicken feed with probiotic bacteria capable of inhibiting *C. jejuni* and/or administering *Campylobacter*-specific phages to chickens [22,23,24]. *Campylobacter* phages are natural “predators” that could potentially be able to bring *C. jejuni* under control. Phages are already being used to control other food-borne pathogens, such as *Salmonella* spec. and *Listeria monocytogenes* [25,26]. However, despite their historical use in typing *Campylobacter* spec., there are limited number of reports about application of *Campylobacter* phages for therapeutic purposes [27,28].

The goal of this review is to analyze the results shown by research using phage-based applications targeting *C. jejuni* and summarize the scientific and regulatory experience from similar phage products against other food-borne pathogens.

## 2. Classification of *C. jejuni* Phages and Receptor Specificity

Phage typing has been used extensively for epidemiology studies of *C. jejuni* and *Campylobacter coli* [29]. The number of *C. jejuni* bacteriophages reported to date exceeds 170 and the majority of these phages exhibit a narrow host range [13]. Most *Campylobacter* phages are lytic, i.e., virulent or able to lyse the host bacteria upon infection and are thus preferred due to their immediate effect [30]. Lysogenic or temperate phages are generally avoided for their ability to integrate into a bacterial genome and transduce virulence genes from strain to strain [31,32,33,34]. There are several ways to classify phages. Generally, phage classification schemes are based on their morphology, genetic makeup (e.g., DNA vs. RNA) or genome size (Table 1) [35]. Morphologically, the majority of *C. jejuni* phages are categorized into the family Myoviridae (Bradley’s morphotype A1, contractile tail), while some have been designated as Siphoviridae (Bradley’s morphotype B1, non-contractile tail) [36].

Lytic *Campylobacter* phages are divided into three groups according to the size of their genome: group I contains relatively rare phages with large genomes (approx. 320–425 kbp), while groups II and III include most *Campylobacter* phages isolated and characterized to date [37]. These are all distantly related to T4 bacteriophages and have smaller genomes: 175 to 183 kbp (group II) and 131 to 135 kbp (group III). Group II and III Campylophages exhibit resistance to many restriction endonucleases. These phages are characterized with great similarities within each group but differences between the groups [37,38]. Group II Campylophages are often specific to both *C. jejuni* and *C. coli*, while group III phages demonstrate specificity exclusively to *C. jejuni* and have stronger lytic activity [37].

The latest phage classification scheme, which is based on whole genome sequencing and protein homology, puts group II and III phages into the Eucampyvirinae sub-family of myoviridae as “CP220-like viruses” and “CP8-unalike viruses”/“CP8 virus” [37,38,39]. Sørensen and colleagues showed, with some exceptions, the connection between receptor dependency and grouping of *C. jejuni* phages. For example, the majority of group III phages use capsular polysaccharide (CPS) receptors, while group II phages target mostly flagellar receptors [27,40].

## 3. Phage Applications Targeting Food-Borne Pathogens

The Earth’s biosphere is estimated to contain approximately 10^32^ phage particles, exceeding the number of prokaryotic organisms by 10-fold [33,41,42]. Bacteriophages differ in size and shape, and are characterized with incredible genome mosaicity resulting from different environmental pressures, including the adaptation to continuous bacterial resistance [43]. At the time of writing this article, only 8852 complete genomic sequences have been listed in the NCBI database of phage genomes. Thus, numerous novel phage genes, potentially leading to development of new phage-based drugs, are yet to be discovered and studied.

Bacteriophages were successfully used for human therapy almost immediately after their discovery in the beginning of the 20th century [41]. After the triumph of penicillin in 1940s, phage use gradually dwindled in the West, but continued in Georgia (the former USSR) and Poland [44]. The Eliava Institute of Bacteriophage, Microbiology and Virology founded by George Eliava and d’Herelle in Tbilisi in the 1930s still continues its work and has accumulated extensive experience in phage work and phage therapy over the decades. The advent of the era of multi-drug resistant bacteria has rekindled the interest in bacteriophages due to several major advantages of phages over antibiotics. First, bacteriophages are effective regardless of the susceptibility of their host bacteria to antibiotics, i.e., antibiotic-resistant bacterial strains are affected in the same way and with the same effectiveness as non-resistant strains. Second, phages are ubiquitous and relatively easy to isolate. Third, their narrow specificity is instrumental in avoiding the disruption of the entire microbiome in the treatment subject [45]. The fourth advantage is that new, mutant, phages may be developed in a matter of days [46]. Phage-based applications can be used as an alternative to antibiotics in live animals, in situ on processed meat and other foods, and for decontamination of surfaces and equipment in food processing to ensure that no multi-drug resistant bacterial strains selected as a result of antibiotic-based evolutionary pressure are released into the wastewater and thus into the environment [25]. Thus, bacteriophage application is manifold and covers different stages of primary (live animals) and secondary (harvested meats) production in animal farm settings. Having no associated toxicity, phage treatments can be complexed with other methods, such as probiotic supplements, aimed at reduction of *C. jejuni* [1,4,45].

Humans have been continuously exposed to phages via drinking water and fresh foods without any adverse reaction ever recorded [24]. In fact, phages have been isolated from human saliva and intestines [47]. A gram of human feces contains approximately 10^8^–10^9^ virus-like particles comprised mostly of DNA phages [48,49]. Based on the evaluation of experts qualified by scientific training and experience, the United States Food and Drug Administration (FDA) recognized phages and phage derivatives as generally regarded as safe (GRAS) through the 1958 Food Additives Amendment to the Federal Food, Drug, and Cosmetic Act [50,51]. A phage manufacturer, however, has to provide proof to the FDA in the form of a GRAS notification indicating the intended use [26]. For example, the GRAS notification for ShigaShield^TM^—a phage product developed by Intralytix against Shigella—specifically states that the GRAS status has been determined “through scientific procedures” [52].

Although most phages are harmless to human health and environment, not all phages are safe, as some lysogenic phages can actually carry and transduce virulence genes. For example, *E. coli* O157:H7 and *Streptococcus pyogenes* owe their virulence to the acquisition of phage virulence genes into their genomes [33]. To avoid “phage lysogenic conversion”, i.e., changing the properties of specific bacteria, lytic phages must be used, as they immediately lyse the host and do not integrate into the bacterial host genome. For assuring the safety of phage preparations used in animal and human therapy, the latest achievements in sequencing technology are utilized. Bioinformatics methods can rapidly predict undesirable phage properties. The GRAS designation and the FDA approval of phages P100 and LMP-102 were specifically based on this approach following phage sequencing.

Advances in phage research have increased the interest in their possible use as biocontrol agents to preserve or decontaminate food products and, thus, prolong their shelf life while preventing frequent disease outbreaks. According to the US CDC, 841 foodborne disease outbreaks were reported by 50 states, Washington, D.C., and Puerto Rico, resulting in 14,481 illnesses, 827 hospitalizations, 20 deaths, and 14 food recalls in 2017 [53]. The major foodborne bacterial pathogens most frequently associated with lethal outcomes are *L. monocytogenes*, *S. enterica* and *E. coli* (e.g., *E. coli* O157:H7), followed by *C. jejuni* [26,54]. Naturally, these food-borne pathogens became the focus of the first phage-based applications developed. In dairy production, for example, *S. enterica* contamination can occur at virtually any stage and phages can be used to reduce the shedding of *Salmonella* in farm animals, as a food additive in post-harvest food products, and as biocontrol agents in food processing to control the pathogen [25]. Phage therapy against *Salmonella* has been successful in broiler production and significantly reduced the pathogen in the ceca of broilers [25].

The general recognition of phage applications as GRAS also prepared the ground for regulatory clearance of several phage-based biocontrol products targeting *L. monocytogenes*, *S. enterica* and *E. coli*. The first such product, ListShield^TM^ (LMP-102) from Intralytix targeting *L. monocytogenes* “ready to eat” (RTE) foods, was approved by the US FDA in 2006. A similar biocontrol agent against *L. monocytogenes*, Listex P100, got the FDA approval for the use in meat, cheese and other foods, including fish, shellfish, fresh fruits and vegetables. Thereupon ListShield^TM^ was approved in Canada and Israel, while Listex has been approved in Switzerland for use in cheesemaking. Approvals for further phage applications followed: in 2007, the FDA approved a product from Omnilytics for decontamination of live animals from *E. coli* and *Salmonella*. Within the same year, the FDA followed up with the approval of new phage applications, including the “Finalyze” spray against *E. coli* O157:H7 in cattle (Elanco Food Solutions) and “Armament” against *Salmonella* in poultry. Another phage application from Intralytix-EcoShield, 95–100% effective against *E. coli* O157:H7, received the FDA’s regulatory approval in 2011 for use on red meats prior to grinding [30].

Up to the present time, the FDA’s GRAS inventory includes phage applications against Shiga-like toxin-producing *E. coli*, *L. monocytogenes*, *Salmonella* and *Shigella*. None of the listed products target *Campylobacter* spp., although Intralytix and Micreos have publicly expressed their interest in pursuing the development of such applications [24].

To date, there are only very few patents on phage products relating to their use for *Campylobacter* germ load reduction in poultry flocks or on processed meat.

Instead of a complete bacteriophage, Fischetti and colleagues patented a process that uses a bacteriophage-derived lysis enzyme for bacterial decontamination of food products [55]. This bacteriophage-derived lysis enzyme can be increased in its effectiveness of bacteriolysis by modification, e.g., by construction of chimeric lytic enzymes, shuffled lytic enzymes or by additional holin proteins. This patent also includes *Campylobacter* spp. in the list of potential target organisms. However, a specific bacteriophage-derived lysis enzyme was not named in the 2002 patent [56]. In the 2004 version, the bacteriophage-derived lysis enzyme PaI with activity against *Streptococcus pneumoniae* and other streptococci of the Viridans group were also listed [55].

Based on their in vivo study, Connerton patented the application of the bacteriophages CP8 and CP34 [2] to reduce *Campylobacter* spp. in the intestine of birds [57] (for further details see Section 4.1, study II and III, as well as Table 2).

Burnett and coworkers patented a procedure to reduce or to prevent bacterial contamination of any type of food product using bacteriophages [58]. The bacteriophages are to be administered in a specific embodiment, which may contain a buffer, surfactant, adjuvants and enhancers to prevent degradation of the bacteriophage and even to enhance its performance as an antibacterial agent. Application of the bacteriophages can also be done on non-food surfaces or water systems. Besides the shown example, *L. monocytogenes*, *Campylobacter* spp. and other bacteria are listed as possible target organisms for this procedure.

The patent of Ter Haar and Hanna complements the application of bacteriophages to food by a rubbing process using a vibratory conveyor to distribute the bacteriophage more efficiently on the food surface [59].

The European Food Safety Agency (EFSA) approached the usage of phages as biocontrol agents rather carefully and issued three scientific opinions in 2009, 2012 and 2016. While the 2009 opinion focused on the nature of bacteriophages and their potential use for decontamination of foods, the 2012 opinion was a response to Micreos’ application for the approval of LISTEX^TM^ P100 to reduce *L. monocytogenes* [60]. This statement addressed safety and efficacy issues of the phage product, but concluded that the product was safe, although the agency had several concerns, specifically on (i) the absence of industrial scale studies, (ii) the limited number of phages used, and (iii) the lack of evidence of significant pathogen reduction. The 2016 opinion finally recognized the safety and efficacy of LISTEX^TM^ P100 for the use on meat, poultry, fish, shellfish and dairy products at up to 1 × 10^9^ PFU [60].

## 4. Studies Reveal Variable Specificity and Efficacy of *C. jejuni* Phages

Studies from different countries resulted in 0.2–3 log_10_ levels of reduction of *C. jejuni* post phage treatment depending on the phage(s), the host strain used, and the method of application. The efficacy data from these studies indicate that understanding phage–host interactions is an important prerequisite to discovering the underlying mechanisms of phage specificity [27]. Studies of *Campylobacter*-phages are listed below in a chronological order.

### 4.1. In Vivo Testing

In 2005, Wagenaar et al. (study I) tested monophages and two-phage applications against *C. jejuni* in young broilers for preventive and therapeutic purposes. By day 5, the bacterial CFU counts were up again after initial 3 log_10_ level reduction and plateaued at 1 log_10_ level below the control group. The same was observed with the preventive group following delayed colonization of birds by one week. The efficacy of a polyphage application was comparable to that of individual phages used in vitro. This study also demonstrated that polyphages could slow or prevent development of phage-resistant mutants [61].

Similarly, Loc Carrillo and colleagues (study II) used two *Campylobacter* phages (CP8 and CP34) in 25-day old broilers applied orally in combination with an antacid. Phage CP8 was active against one tested *C. jejuni* strain (GIIC8), but not against the other tested *C. jejuni* strain (HPC5), despite its in vitro susceptibility to the same phage. In contrast, phage CP34 reduced both *C. jejuni* test strains HPC5 and GIIC8 and maintained the reduction for 5 days. However, 72 h post treatment counts began to increase indicating that in vitro efficacy may be drastically different from the in vivo results [2].

El-Shibiny et al. (study III) applied the same phages (CP8 and CP34) to birds pre-colonized with a wild type *C. coli* isolate. In this study, another phage, CP220, was also tested in chicks pre-infected at 20 days old with *C*. *jejuni* strain HPC5. The most important finding of this study was that only high doses of phages (10^9^ PFU) were able to reduce cecal counts of *C. coli* by 1–2 log_10_ levels within 48 h and a reduction of 2.1 log_10_ levels of *C. jejuni* HPC5 could be obtained 24 h post phage (CP220) administration [62].

Carvalho and colleagues (study IV) also tested a two-phage cocktail (with different lytic spectra) in birds that were colonized by *C. jejuni* and *C*. *coli.* Better results were produced when phages were given to chickens with feed, compared to oral gavage. However, 6% of the *Campylobacter* isolates were resistant upon passage through the bird guts [63].

In contrast, Kittler and coworkers (study V) tested a phage cocktail of four group III phages in three commercial broiler farms (in one farm, penicillin was applied over 3 days in advance). Each setup included an experimental group and a control group. One of the experimental groups showed significant reduction of *C. jejuni* counts below the detection limit on day 1 and still over 3 log_10_ levels on the day of slaughter, demonstrating that maximum reduction was achievable 1–4 days prior to slaughter [64]. The fact that sometimes genotypically identical *C. jejuni* strains within flocks exhibited different phage susceptibility profiles and varying biotypes was an important observation linking the lack of efficacy of phage applications with genotypic variability of *C. jejuni* isolates.

All five studies were similar in their goal to determine the efficacy of phages, isolated from the environment of commercial broiler farms or obtained from bio banks, in reduction of *C. jejuni* and/or *C. coli* in artificially (studies I–IV) and naturally (study V) infected broilers. However, fundamentally different approaches could be observed from study to study. First, a good colonizing strain of *C*. *jejuni* and phages that were previously characterized [36] were used in study I. Because neither the bacterial strain, nor the phages were naturally isolated from local chicken meat or excreta, this model cannot be considered as suitable for wider applications of phages for preventive or therapeutic purposes in farms. Still, this was also the first in vivo study to demonstrate that application of a high dose (10^11^ PFU/mL) of phages to live chickens did not lead to any signs of pathology and resulted in considerable reduction of the host strain 24 h post application. Phage 71 (a group III lytic phage), which was used in this study, was able to achieve 1 log_10_ level reduction that was maintained within the experimental flock over 30 days and up to 3 log_10_ levels reduction of *C*. *jejuni* C356 in the following 24 h. Another important observation made in this study was that phage 69, which was less potent and belonged to the same group (group III) of phages, boosted the potency of phage 71, when administered together. The long-term effect of applying the two phages together was 1.5 log_10_ level reduction of the host bacteria compared with 1 log_10_ level when phage 71 was used alone.

Studies II and III tested the same two (CP8 and CP34) phages separately, as well as in a cocktail, against *C*. *jejuni* and *C*. *coli,* respectively. Additionally, study III also tested a group II phage CP220 against *C*. *jejuni* and demonstrated the immediate efficacy of a high dose of phages. The benefit of application of phages as a cocktail had already been observed in study I. Therefore, the subsequent studies (III and IV) started administering phages to the chicks in this manner. The major drawback of most studies (I, III and V) was to use phages obtained elsewhere. Local isolates were only used in studies II and IV. These studies used phage isolates from farms around the UK (study II) and free-range chickens (study IV, although without indicating the precise geographical origin, which, presumably, was Portugal). Additionally, study V used strain NCTC 12662 to test the efficacy of phages, while evidence that the three experimental sites were infected with different biotypes of *C. jejuni* probably already existed. In our opinion, it would be better to use *C*. *jejuni* isolates from each respective group in this study (V), rather than strain NCTC 12662.

Fundamental differences were observed in the method of phage selection in two out of five studies. For phage selection Study II used only *C*. *jejuni* (host derived isolates from the same sample or the laboratory strain NCTC 12662), while study IV utilized both *C*. *jejuni* and *C*. *coli*. This turned out to be beneficial for study IV and resulted in selection for phages that were active against both *C*. *coli* and *C*. *jejuni* exhibiting synergistic efficacy. In contrast, in study II exclusively, CP8-like, i.e., CPS-specific, phages were selected. Another drawback of study II was to use only oral gavage as the route of administration, which seems impractical in large-scale operations, should phage therapy ever be approved for broiler farm use. Oral gavage in study IV was compared with mixing the phage cocktail in the chicken feed, which proved to be more effective. Although it is hard to estimate how the chicken feed contributed to better stability of phage preparations, the results showed that this route of administration led to more sustainable reduction of *Campylobacter.* However, this effect could also be achieved due to the use of specifically different phages (group II versus group III). In study V, drinking water was used as the means of phage delivery, which certainly is more practical than oral gavage in large-scale operations. However, in the course of the study, no stability experiments were conducted to demonstrate survival of phages in drinking water. Study II, for example, conducted such stability experiments before mixing the phages with calcium carbonate.

In terms of phage resistance, the data from the five studies also differ. For example, studies II and III identified resistance occurring at a rate <4% and 2%, respectively. In contrast, 6% resistance, which was not explained well, was observed in study IV already prior to administering the phages. Post administration resistance in this study doubled and reached 13%. Study V reported no effect of phages in 2 out of the 3 experimental groups, which could be due to resistance or because of the use of strain NCTC 12662 instead of a local *C*. *jejuni* isolate for demonstration of efficacy. Moreover, while the second study reported that the resistant strains were compromised in their ability to colonize chickens, rapidly reverting to the phage sensitive form, study IV did not observe this phenomenon, while study V identified multiple biotypes of *C*. *jejuni* in all three experimental sites. At the same time, all five studies reported the use of phages of approximately the same genomic size and morphology, placing them into the Myoviridae family. From the genomic sizes of the phages reported by study IV, it can be understood that the phages belonged to group II, which tend to be mostly flagellotropic. This can also be deduced from the fact that the phages described in the publication were specific to both *C. jejuni* and *C*. *coli*. Study III reported the use of phages from both groups (II and III) and concluded that application of cocktails using phages from different groups can increase the potency of phage cocktails.

Regarding the use of *C*. *jejuni* strains for artificial infections, in the first three studies, the scientists used chickens experimentally infected with a single *C. jejuni* or *C. coli* strain. In reality, research suggests that multiple strains of this pathogen may be present even in a single individual bird. Still, the results demonstrated that the two phages of similar size from the same Myoviridae family may have quite different lytic profiles and efficacy. This supports evidence that local phage isolates could be more effective towards the host bacterial strains isolated in the same locality, although it is not clear whether the phage isolates in study II were not obtained as a result of selection using *C*. *jejuni* strain NCTC 12662. The two phages clearly exhibited different efficacy profiles towards the two bacterial hosts, *C. jejuni* HPC5 and *C. jejuni* GIIC8, and were used as single agents. Administered as a cocktail, these phages could demonstrate improved efficacy against a single host as well as against a combination of the two hosts used.

Overall, the results from all the five in vivo studies agree that phages can effectively reduce *C. jejuni* if administered 24–48 h prior to slaughter at high M.O.I. experimentally established prior to administering. Efficacy of phage applications may be improved if local isolates of both bacteriophages and putative hosts are used. Further studies are needed to substantiate these findings and to achieve reduction of *C*. *jejuni* exceeding 2 log_10_ levels, which, according to the authors of studies II and IV and based on mathematical models, can be translated into an approximately 30-fold reduction of the pathogen. This, however, is not nearly enough to achieve sustainable public benefit. Therefore, further research should probably focus on using cocktails consisting of both CPS-specific (group III) and flagellotropic (group II) phages applied as a cocktail with carefully planned stability experiments, proving that phage preparations can withstand the conditions of the selected route of administration, as well as that of a buffer or carrier substance.

### 4.2. In Situ Testing

Five research teams addressed the question of whether application of phages at refrigerated temperatures could be effective against *C*. *jejuni*.

Reduction of this pathogen was demonstrated by two different groups—Atterbury et al. (study I) and Goode et al. (study II) [15,65]—on artificially contaminated skin following treatment with high titers of group III lytic phages.

Orquera et al. (study III), who also used group II phages in their research, concluded that the latter were unable to reduce *C. jejuni* in situ at 4 °C [66]. Similar to studies I and II, Orquera and colleagues obtained the phages (CP81 and NCTC 12684) and the host bacterial strains (*C*. *jejuni* NCTC 11168 and *C*. *coli* NCTC 12668) from bio banks. The experiments were conducted at 37 °C in broth, as well as at 4 °C on meat. While 1–2 log_10_ levels reduction was achieved in broth, no significant change in the numbers of *C*. *jejuni* was reported at 4 °C on meat. In contrast, phages against *Y. enterocolitica* were able to reduce the host bacteria up to 2 log_10_ levels at the same temperature on meat in the same study. These results are consistent with the evidence that *C*. *jejuni* may remain viable at 4 °C, although it cannot multiply at this temperature. This could be the reason why phage treatments are less effective at refrigerated temperatures.

The conclusions from studies I and II also contradict those of Firlieyanti et al. (study IV) and Zampara et al. (study V). In their experiments, the latter two teams used either group II flagellotropic phages or phage preparations consisting of group II and group III bacteriophages. Reduction of *C. jejuni* at low temperatures in these studies was found to be modest due to poor efficacy of group II flagellotropic phages at such temperatures, according to the authors of the published results [27,67].

As mentioned earlier, in studies I–III the host bacteria and their bacteriophages were obtained from bio banks and were not isolated from any specific locality. For example, *C*. *jejuni* NCTC 12662 and *C*. *jejuni* C222 were used as the host strains in studies I and II, respectively. These studies, therefore, cannot represent a realistic scenario where various types of host bacteria and their phages exist. Nevertheless, they provided valuable information that, if applied at high titers, some phages are indeed able to significantly reduce *C*. *jejuni* in situ. The success of the first study, however, could be explained by the fact that *C*. *jejuni* NCTC 12662 is readily infected with phages and, for this reason, is often used for selection of bacteriophages. That is obviously why study IV used this strain for phage propagation purposes.

Study IV used the isolates of the host bacterial strains and their phages from the same local source (specifically, chicken livers). The results, as seen in Table 2, were modest. It could be argued that studies I and II were more successful in reduction of *C*. *jejuni* because both teams used group III lytic phages. However, phage CP81 used in the study III was also a group III phage, but inefficient with the particular strain of *C. jejuni* (NCTC 11162). Indeed, this strain could be less susceptible to phages compared to *C. jejuni* NCTC 11162 used in previous research. Thus, wherever group III phages were utilized in studies III and IV, the low rate of clearance of *C. jejuni* can be explained because the efficacy studies were conducted using host strains other than NCTC 12662.

Study IV effectively demonstrated that *C*. *jejuni* could remain viable at 4 °C during a 72 h period, which is critical for phage survival. Viability was confirmed by unchanged levels of phage recovery from such samples. However, because the pathogen does not actively divide at such low temperatures, phages also remain static. Another important observation of study IV was that flagellotropic phages Φ3 and Φ15 were able to lyse three out of five chicken liver isolates of *C*. *jejuni* in addition to the control strain HPC5. This indicates that the outcome of phage treatments could be largely dependent upon the origin of both bacteriophages and the host bacteria.

Study V conducted by Zampara and colleagues investigated the efficacy of 19 *Campylobacter* phages isolated locally throughout chicken farms in Denmark and used two *C*. *jejuni* strains, NCTC 12662 and RM1221. In particular, since previous research focused mostly on group III bacteriophages, this team also included group II (mostly flagellotropic) phages in their experiments and found significant difference between the ability of group III and group II phages to reduce *C*. *jejuni* loads. Specifically, group III phages led to 0.55 log_10_ level reduction at 4 °C on chicken skin, while no reduction was achieved by group II phages. To answer the question of whether decreased motility could be the reason for the low efficacy displayed by flagellotropic phages, the RM1221 strain was also used as a control. This *C*. *jejuni* strain is characterized by enhanced motility. Decreased motility of NCTC 12662 at 4 °C was not observed. Therefore, the inefficiency of group II phages could not be explained by decreased motility of the host bacteria. The noteworthy observation of study V was that, despite the use of NCTC 12662 (the strain that is readily infected with most phages), significantly lower efficacy results were obtained with group III phages in this study.

As was the case with the in vivo research discussed above, the five in situ studies evolved sequentially. The first two studies introduced the idea of phage application at refrigerated temperatures, while the subsequent investigations tried to answer more specific questions or confirm the findings of the previous research. Thus, studies IV and V evolved into more thorough investigations utilizing a more diverse population of phages. Future research, therefore, should probably narrow down on specific phage isolates and screen those against multiple isolates of *C*. *jejuni*. However, both should come from the same source of meats and chicken excreta. Again, cocktails consisting of different phage isolates should be used. This time more pertinent questions must be answered, such as: could in vivo treatment be complemented by in situ treatment of the same chicken meat post slaughter with the same, or different, phages or phage cocktails?

## 5. The Future of Campylophages: Improving Selection and Handling Resistance

### 5.1. Improving Phage Selection

An improved method of isolation of *Campylobacter* phages was suggested by An-Chi Tsuei and colleagues by pooling the parts of the whole chicken rinses. The method allowed for 28% recovery of *C. jejuni* phages [68]. This approach is consistent with the isolation of *C. jejuni* from retail chicken by rinsing bird carcasses with buffered peptone water [69]. Phage isolation, however does not pose a major problem. What needs to be addressed in research studies is using local isolates of *C*. *jejuni* for selection of phages to increase their diversity. For example, for enrichment of phage isolates, Furuta et al., who found that phage susceptibility profiles of the Japanese *Campylobacter* strains were different from those of non-Japanese ones, suggest using locally isolated *C. jejuni* strains [70]. Previous research of *Campylobacter* phages relied heavily on just a few *C*. *jejuni* strains, notably NCTC 12662, which was frequently used for propagation and enrichment of phages, as discussed previously in this work. However, it is essential to use a broad range of strains for the selection and isolation of *Campylobacter* phages, because the primary phage isolation strain largely determines both phage type (CP81 or CP220) and receptors (CPS or flagella), as demonstrated by Sørensen and colleagues [40]. The use of control *C. jejuni* strains, also known as phage propagating strains (NCTC 12662 (PT14), NCTC 12660, NCTC 12661 and NCTC 12664), susceptible to a wide range of phages may have resulted in biased selection of the latter [37,71]. For example, Sørensen and colleagues used *C*. *jejuni* strains with different CPS profiles to isolate bacteriophages from free-range poultry samples. All 20 phage isolates were obtained using *C*. *jejuni* NCTC 12658, NCTC 12662 and RM1221. The phage isolates were subject to host range and genome restriction profile determination analysis concluding that the host range and morphology of the isolated phages correlated with the bacterial strains used for isolation. Thus, NCTC 12662 allowed for the isolation of CP81-type phages, while RM1221 determined CP220-type selection, as shown by phage specificity studies using either acapsular or non-motile mutants [72]. Thus, to diversify a phage collection, local strains of *C*. *jejuni* must be isolated and characterized first.

### 5.2. Understanding and Handling Phage Resistance in C. jejuni

Bacterial resistance to phages was first noted by d’Herelle [45]. Later Delbrück and Luria also observed that initial lysis of bacteria resulted in subsequent re-growth [73]. Today it is widely known that resistance of bacterial strains to bacteriophages can be a major drawback in the development of therapeutic phage applications. There is also some concern that development of phage resistant bacterial strains could lead to limited long-term sustainability of phage applications, as it is the case now with antibiotic drugs [74]. Resistance of *C*. *jejuni* to their bacteriophages is making the wider use of the latter challenging. Development of phage resistance was noted in most *Campylobacter* studies, both in vivo and in vitro. For example, in their 2005 in vivo study Loc Carrillo and colleagues identified 4% and 11% resistant isolates of *C. jejuni* HPC5 [2]. The majority of these resistant isolates (97%) reverted back to the phage sensitive phenotype when tested again later, as confirmed by MLST analysis [2]. Notably, this study also showed that different phages of the same group (lytic group III phages) can elicit different levels of resistance in the host bacterial strain. Thus, phage CP8 was more effective in infecting and reducing HPC5 compared to phage CP34 and with the majority of *C*. *jejuni* subpopulations remaining phage-sensitive. The ability of phage resistant subpopulations of *C*. *jejuni* to revert back to phage-sensitive form was further observed by Carvalho and colleagues, who noted that colonization of chicken gut was possible via quick reversion of the resistant phenotype of *C*. *jejuni* to the original parental phenotype [63].

Analysis of previous research identified at least three distinct mechanisms of phage resistance in *C*. *jejuni*. Additionally, all bacteria have restriction modification systems to recognize and neutralize foreign DNA. The mechanisms of bacterial resistance to bacteriophage infection are as follows:(a)Receptor modification preventing phage adsorption(b)Diversity of *C. jejuni* strains(c)Resistance due to spontaneous mutations(d)Intracellular degradation of phage DNA(e)CRISPR-Cas system mediated bacterial dormancy

#### 5.2.1. Receptor Modification Preventing Phage Adsorption

Receptor modification is the most common mechanism of resistance to phages in bacteria. Such modifications may arise due to point mutations of receptor-encoding genes or result from changes in their expression [33].

Recently, the phase variable *O*-methyl phosphoramidate (MeOPN) moiety of the CPS has been shown to be the determinant of the resistance in *C*. *jejuni* NCTC 11168R. Experiments revealed that resistance can be acquired in vivo and a particular strain can transfer cross-resistance to other phages. Thus, phase variable CPS structures are able to modulate phage infectivity of *C. jejuni*. Changes in these structures arise in response to the constant exposure to phages in the chicken gut resulting in continuous and dynamic phage–host co-evolution [42,72].

In the context of group II flagellotropic phages, mutation of flagellar receptors could also play a role in phage resistance. That is what Lis and Connerton reported in their study of a *flaB* knockout strain of *C. jejuni* PT14. According to their research, *flaB* inactivation led to a decrease in swarming motility of bacterial cells and increased their susceptibility of this strain to phage CP_F1. Thus, maintenance of *flaB* gene, according to this study, could be an evolutionary adaptation towards increasing the diversity of antigens in order to evade infection from flagellotropic phages [75].

#### 5.2.2. Diversity of *C. jejuni* Strains

The extreme diversity of *C. jejuni* strains is one of the causes of development of resistance. Research showed that several *C. jejuni* strains may coexist in one broiler flock at any given time [21,63,64]. Co-existence of strains with different phage susceptibilities in broiler flocks was observed by other authors and explains the finding of Scott et al. and Connerton et al. that succession of *C*. *jejuni* strains during phage application occurs due to new genotypes instead of the existing strains becoming resistant [76,77]. The reason why *C. jejuni* CFU counts rise again after the initial reduction post phage application is most probably caused by the susceptible strains giving way to non-susceptible ones. For example, the diversity of *C*. *jejuni* strains was confirmed by Kittler and colleagues by the use of MLST analysis to characterize *C. jejuni* isolates from three different farms. They identified two sequence types (STs) in each of the two different experimental groups. The sequence types and phage susceptibility did not coincide from group to group. Moreover, in two out of four total possible biotypes determined among the isolates, these different biotypes sometimes belonged to one sequence type, thus making the results between the three trials highly variable. For example, in trial 3, the susceptibility and biotype matched, while in trial 1 reduction took place in vivo, despite no susceptibility being shown in vitro [64]. Previous studies also observed that in vitro and in vivo data may differ substantially, resulting from different genetic variants of *Campylobacter* in different flocks [78].

#### 5.2.3. Resistance Due to Spontaneous Mutations

Phage-resistant bacterial strains sometimes arise due to spontaneous mutations. Once a bacteriophage threat is eliminated, bacteria revert to a phage-sensitive strain [35]. Loc Carrillo et al. and Scott et al. showed that phage resistant isolates may arise due to chromosomal inversion and that resistant strains exhibit decreased ability to colonize birds [2,77]. In fact, phage-susceptible *C*. *jejuni* subpopulations can overgrow phage resistant populations due to increased motility, as shown by Kittler et al., although only in vitro [64]. Brathwaite and colleagues, also noticed the decreased ability of biofilm-associated *C. jejuni* subpopulation, infected with lytic phages in CSLC, to colonize broilers. RNA-seq analysis found that the decreased fitness was due to down-regulation of important stress-response genes, including the major flagellar protein FlaA [79]. The conclusion drawn from these observations is that, due to the short lifespan of broilers, it is unclear how the dynamics of resistance continue.

#### 5.2.4. Intracellular Degradation of Phage DNA

Another aspect that has been barely studied so far in the context of phages is the intercellular degradation of extraneous DNA depending on its methylation pattern by restriction modification systems (RM-systems). RM-systems principally consist of three types of subunits: (i) restriction endonuclease subunits (R) facilitating DNA cleavage, (ii) specificity subunits (S) for the detection of specific DNA sequence motifs, and (iii) DNA methylase subunits (M) [80]. A highly conserved methylase that recognizes and methylates the RA^m6^ATTY motif [80] was identified, which significantly increases the transformation rate and was therefore named *Campylobacter* transformation system methyltransferase (*ctsM*) [81]. In turn, reduced RA^m6^ATTY motif methylation leads to increased degradation by methylation-associated RM systems that degrade DNA lacking the correct methylation motif. Consequently, phage resistance may also be due to reduced methylation of specific motifs of phage DNA.

#### 5.2.5. CRISPR-Cas System Mediated Bacterial Dormancy

The innate immunity of bacteria against phages is represented by CRISPR-Cas systems that have been classified into two classes, 6 types, and more than 30 subtypes. Recently it has been observed that, unlike the type I CRISPR-Cas system, which constitutes about 60% of all CRISPR-Cas systems, types III and VI can induce cell dormancy and thus decrease an infected population of cells. It is also thought that the type V CRISPR-Cas system may actually lead to cell death [82].

Watson et al. demonstrated in their study that infection of *Pectobacterium atrosepticum* by two independent phages, ΦTE and ΦM1, activates type I CRISPR immunity. This results in abortive infection and leads to death of infected cells. Therewith the phage propagation was decreased. Thus CRISPR-Cas type I systems may serve as tool to reduce phage infection and protect given bacterial populations from infection [82].

#### 5.2.6. Handling Resistance

Two possible solutions exist to control resistance to phages in farm environments. The first is to apply phages strictly for 24 to 48 h to birds designated for slaughter after physical sequestration in order to prevent the spread of the infection to other birds. The in vivo studies discussed earlier identified that the efficacy of the high titer phage applications can be observed within 48 h, after which it either drops entirely or is maintained at a lower level. Therefore, application of phages towards the end of a production cycle (before slaughter and possibly on carcasses) could alleviate selective pressure on the pathogen.

Another solution is modification of phage products. As mentioned earlier, using phage cocktails consisting of two or more phages, preferably from different groups, is more effective than monophage applications. Hammerl and colleagues were able to demonstrate this in using group III and group II phages in a pre-harvest treatment showing reduction up to 3 log_10_ levels of fecal counts of *Campylobacter* in 20-day-old chickens administered successively with the two phages.. While the group III phage alone was not effective, a combination of the two (group II and III) proved to be efficient [83]. The advantage of using applications consisting of multiple phages is that bacterial resistance comes at a certain fitness cost. Such a fitness cost can be maximized by simultaneous pressure from multiphage applications. In response, the pressure exerted on bacteria from combinations of phages would have to be met with multiple mutations in a given bacterial cell, which means that resistant bacteria would possess less cellular fitness and could eventually be outgrown by a non-resistant phenotype [74]. This was exactly the case with resistant *C*. *jejuni* phenotypes in some studies discussed earlier. Finally, avoiding application of monophages decreases development of cross-resistance when, for example, different phages may target the same bacterial surface receptor. The efficacy of bacteriophage cocktails, thus, could be increased by alternation. Additionally, specific phages within cocktails could be replaced with new characterized phages from an existing library or with new phage isolates from the environment. In any case, preventing resistance requires constant surveillance and obtaining new phage isolates. This makes isolation and characterization of wild type phages extremely important in an effort to obtain potent isolates and prevent emergence of phage resistant phenotypes.

Additionally, the third effective route to control phage resistance in bacteria could be genetic modification of existing phages. However, the regulatory path of such phage preparations is presently unclear.

## 6. Determining Safety of Phages in the Era of “Omics”

Despite the fact that bacteriophages are the most ubiquitous organisms inhabiting every niche of the biosphere, there is still an incredible lack of both genotypic and phenotypic information about these viruses largely due to traditional culture-based approach of their identification. However, recent rapid advances in sequencing technologies have revolutionized the ability to produce high-throughput sequencing data of whole bacterial and viral genomes without the need for their prior isolation, as in the technique of metagenomics. At the same time, the cost of sequencing has decreased considerably. Metagenomics (genomic analysis of a pool or a community of organisms), single cell genomics (analysis of the genome of a single cell) and other “omics” technologies, together with bioinformatics and statistical analysis tools, have created a powerful basis for the analysis of large sequencing datasets and shed light on bacteriophages and their interaction with their host bacteria.

There are several high throughput whole genome sequencing technologies available today, each with their own advantages and disadvantages, such as the read length and the sequencing error rate. Hybrid sequencing can be used in cases where low error rate and large read length are required [84]. Bioinformatics tools available for the assembly, mapping and identification of bacteriophage genomes from the perspective of determining their safety and thus applicability in human or animal therapy have, consequently, been evolving rapidly.

Although phages are generally considered safe, the need for guidelines and methods to assure the safety of their genomes still exists. Lysogenic phages often carry undesirable genes that could be detrimental to the bacterial host, upon expression of virulence factors, or proteins able to degrade various antibiotics. Therefore, guidelines are needed to analyze and differentiate phages designated for therapy in humans or animals by their lifecycle. Such guidelines do not yet exist [85]. The US FDA, for example, relies on emergency protocols proposed by medical doctors for the case-by-case approval of phage cocktails for therapeutic purposes in special “compassionate” use of Investigational New Drugs (IDEs). Recently, a team of scientists from United States Defense Threat Reduction Agency and Biological Defense Research Directorate, Naval Medical Research Center developed a comprehensive workflow to assess the genomic safety of therapeutic bacteriophages [85]. The checkpoints in the proposed assessment include obtaining high quality genomic sequencing data and rigorous analysis of this data to identify sequencing contamination and evaluate safety. The effectiveness of this assessment pipeline has been demonstrated on the bases of two phage genomes and has been proposed by the group as the minimum standard for evaluating the safety of phages. The workflow of data analysis follows these three essential steps: analysis of data quality, determining the lifecycle, and analysis of the proteome. All three stages rely on bioinformatics tools and statistical software for data analysis [85].

The first stage of the safety assessment is data quality: there are several bioinformatics pipelines to evaluate the quality of sequencing data and at least two should be employed in parallel for data QC (quality check) and sequence alignment, including the manual verification of sequence fidelity and definition of the requirements for minimum coverage (in the range from 100× to 400×) of the complete genome. Once this checkpoint is satisfied, the sequence must be mapped to the reference sequence. Again, at this stage, stringency must be defined as 90% or higher if the data can be mapped back to the original genome. The next step is determining the phage lifestyle, i.e., whether a particular phage sequence contains data that could implicate it as having a lysogenic lifestyle. Again, there are bioinformatics options (for example RAST and PHASTER) that can interrogate sequence for the presence of integrase. Presence of the latter, as well as positive prediction of lysogenicity of the proteome of a particular phage (e.g., by the additional bioinformatics tool PHACTS that is based on statistical analysis of the data) will disqualify it from further consideration. Likewise, presence of undesirable elements such as toxins, antibiotic resistance genes and virulence factors may also be checked with bioinformatics tools (for example, ShortBRED can read the sequence in question against databases ARDB and VFDB). Open reading frames can be assessed with tools such as Prodigal and others for homology of predicted ORFs with known proteins existing in various databases by BLASTp, PhAnTOME and other bioinformatics computational methods. The threshold for putative harmful genes could be set to 30% (which is quite low), as opposed to 50–70% for other homologs. Passing all these stringent checkpoints qualifies the phage for potential therapeutic use.

An alternative bioinformatics tool is PhagePhisher, which was developed in Loyola University Chicago and was designed to extract meaningful information from large sequencing datasets [86]. This pipeline processes viral-specific sequences from WGS data based on algorithms that identify non-viral elements by reassembling sequencing data to identify the viruses of interest. Such data can be extracted from sequences of single isolates or complex microbial communities. PhagePhisher was written in python and also encompasses a three-step process: identifying and processing contaminant or host species sequences (referred to as “background genome”) with an option to mask prophage sequences, mapping the viral WGS reads to the background genome, and downstream analysis of those specific reads unable to map to the background, i.e., dissimilar elements. The advantage of this program is that it can work with essentially any mapping or assembly platforms. PhagePhisher was successfully used to identify the *P. aeruginosa* phage (φVader) genome by excluding annotated bacteriophage coding regions from host-derived sequences separated from those of the phage genome (69% of almost a 3 million base pair sequence). The remaining paired-end reads were assembled into the viral genome. Similar to the method discussed above, lysogenic phages can be identified by finding virus-like particles within bacterial genomic sequences because PhagePhisher is able to separate viral elements from bacterial sequences. This approach allowed one to analyze the raw sequencing paired end reads of a WGS survey of the nearshorewaters of Lake Michigan against all publicly available databases of bacterial genomes and plasmids. The paired-end reads were analyzed individually and the ones that mapped to the background were removed, while unmapped reads were then assembled separately. After subsequently BLASTing the five contigs against the nr/nt database without preselecting any particular organism or taxa, all five produced statistically significant hits to various phage sequences. Additionally, by referencing the annotation of the cyanophage genome, one contig was found to be homologous to the phage’s annotated integrase. At the same time, none of the identified contigs showed any homology with bacterial sequences, which proves that this screening can be effectively validated into guidelines for isolation of viral sequences.

There are other similar bioinformatics tools designed to check the quality of phage DNA (identifying potential contamination by host genome), analyze the sequencing results further to determine the lifestyle, and analyze the viral ORFs, for example, CyVerse in the PCPipe available through iVirus project. Besides PHAST, there are ACCLAIM, Prophinder and PhiSpy, which are also available as tools for prophage detection [87].

Finally, SSG (single cell genome) sequencing must be mentioned as a new tool for sequencing single bacterial cells shedding information on prophages contained in the bacterial genome and, in general, on the complex phage–host relationship. Single cells can be isolated by various technologies, such as FACS or confocal laser scanning microscopy. Alongside this, bioinformatics tools also evolve to accommodate these technologies with subsequent analysis, for example, PIPS, a method of classification of prophages-pathogenicity islands [87]. Thus, guidelines for characterizing phages to qualify them for therapeutic use will have to rely on complex methodologies and the validation of these guidelines will most likely be a complex, multi-step process.

## 7. Concluding Remarks

It is obvious that much work needs to be done to achieve significant and consistent reduction of *C. jejuni* at different stages of broiler production. Previously, it was shown using quantitative risk assessment modeling that a 2 log_10_ level reduction of *C. jejuni* in broilers prior to slaughter would significantly decrease the incidence of campylobacteriosis in humans [61]. Application of polyvalent phage preparation is clearly one way to overcome phage resistance, at least partially. Such applications have a successful history of use, for example against *Salmonella*. Phage cocktail treatment significantly reduced this pathogen in ileum and cecal tonsils of chickens [30].

For wider activity and greater potency against *C. jejuni*, it is advisable that polyphage applications (phage cocktails) targeting this pathogen be comprised of the representatives of both CPS-specific and flagellotropic phages or to administer the phages from these groups successively [37,88]. Prevention of recycling of phages reduces resistance rates in a farm environment, therefore conducting phage therapy in birds 1–2 days prior to slaughter could help to avoid resistance [15].

As a final note, due to the ubiquitous presence of *C*. *jejuni* in the environment, successful elimination of this pathogen will most probably be achieved by implementing strict biosecurity measures complexed with phage treatments and probiotic supplements, making continued research in all areas necessary and relevant. This is especially true for *Campylobacter* phages. Characterizing and sequencing new *C. jejuni* phage isolates is much needed to gain more knowledge about these viruses.

## Figures and Tables

**Table 1 animals-10-00279-t001:** Classification of lytic *Campylobacter* Phages.

Group	Phage Size in kbp	Receptor Specificity	Alternative Classification
I	320–425	mostly flagellotropic	–
II	175–183	mostly flagellotropic	“CP220-like viruses”
III	131–135	mostly CPS-specific	“CP8-unalike viruses”

**Table 2 animals-10-00279-t002:** Studies of *C. jejuni*-specific phages listed by year.

Reference	Kind of Study	Phage Group	Phage Origin	*C. jejuni* Strain Used	Results
Atterbury et al., 2003	in situ	Φ2 (NCTC 12674), group III	NCTC	*C. jejuni* NCTC 12662 (PT 14)	Sections of chicken skin, inoculated with different concentrations of *C. jejuni* and bacteriophages, were kept at 4 °C and 20 °C. At maximum phage concentration (10^7^) there was 1.1–1.3 log_10_ level reduction of *C. jejuni* in 4 °C treatment setup and 2.3–2.5 log_10_ reduction in 20 °C treatment setup compared to the controls.
Goode et al., 2003	in situ	NCTC 12673, group III	NCTC	C222	1 log_10_ level reduction observed on chicken skins treated with the phage at the concentration of 10^6^ PFU/cm^2^, compared to the untreated controls.
Wagenaar et al., 2005	in vivo	NCTC 12669, Group III NCTC 12671, Group III	NCTC	C356	After the initial 3 log_10_ levels reduction CFU counts of *C. jejuni* increased again within 5 days and plateaued at 1 log_10_ level lower than control.
Loc Carrillo et al., 2005	in vivo	CP8, Group III CP34, Group III	Retail Chicken	HPC5 GIIC8	0.5–5 log_10_ levels reduction depending on the intestinal site and phage-host combination. The study demonstrated that the greatest reduction was achievable within 24–48 h. Substantial differences were identified between in vitro and in vivo results.
Bigwood et al., 2008	in situ	Cj6 Group not specified	Chicken feces	Chicken isolate	Cooked and raw beef samples inoculated with *C. jejuni* were treated with bacteriophage and stored at 5 °C and 24 °C. The maximum (2 log_10_ levels) reduction was achieved in samples that were treated with high densities of *C. jejuni* and high M.O.I. of the phage at both storage temperatures.
Carvahlo et al., 2010	in vivo	PhiCcoIBB35, Group II PhiCcoIBB37, Group II PhiCcoIBB12, Group II	Free Range Chickens	2140CD1	Approx. 2 log_10_ levels reduction achieved using the cocktail consisting on the three phages. Phage delivery with food was more effective than by oral gavage.
Orquera et al., 2012	in situ	NCTC 12684, group II CP81, group III	NCTC	NCTC 11168	No reduction observed at 4 °C in situ on meat or in vitro. 1 log_10_ reduction was observed in vitro at 37 °C.
Kittler et al., 2013	in vivo	NCTC 12672 NCTC 12673 NCTC 12674 NCTC 12678 All group III phages	NCTC	NCTC 12661 NCTC 12664 NCTC 12660	Phage cocktail was administered to birds via drinking water. Group I: 3.2 log_10_ CFU/g lower *C. jejuni* counts than in the control until slaughter. Group II: No significant reduction Group III: No reduction
Firlieyanti et al., 2016	in situ	Φ3, group II Φ15, group II	Chicken liver isolates	Chicken liver isolates	Modest reduction 0.2 log_10_ level (low C. jejuni inocula)—0.8 log_10_ (high C. jejuni inocula) at 4 °C.
Zampara et al., 2017	in situ	Group II and Group III phages identified previously.	Free range chicken isolates	NCTC 12662 RM 1221	The study concluded that CPS phages bound more tightly to *C. jejuni* compared to flagellotropic phages and, therefore, were more efficient at reducing the pathogen at low temperatures. It was also observed that efficiency of phage cocktails at reducing C. jejuni was higher than that of single phages.

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
