# Peer review of "Application of Campylobacter jejuni Phages: Challenges and Perspectives"

_animals, 2020, doi:10.3390/ani10020279_

Round 1

Reviewer 1 Report

This a very well written and interesting article. However, there are phrasing errors throughout to be checked prior to publication.

Points to consider-

1) In the abstract (line 34) you state that no Campylobacter phage products have been approved. There are patents to some strategies available online, e.g. https://patents.google.com/patent/CA2545018A1/en I recommend commenting on these in the manuscript.

2) Line 70 - explain the term "estimated public benefit" in greater detail.

3) Line 125 - The phrase "environmentally friendly products" needs to be described in full.

4) Table 2 - in the Wagenaar et al section, you use the phrase "went up again". This needs to be changed to rephrased.

5) Table 2 - you alternate between log and log 10 in different sections. This needs to be clarified.

6) Line 285 - rephrase "bounce back".

Author Response

Reviewer I

1) In the abstract (line 34) you state that no Campylobacter phage products have been approved. There are patents to some strategies available online, e.g. https://patents.google.com/patent/CA2545018A1/en I recommend commenting on these in the manuscript.

Answer: Because of the very limited number of words we have corrected our statement in the abstract to “however only very few Campylobacter phage product has been approved anywhere to date”; and several patents have been added to section 3. However, there seems to be no definition in Animals' citation style for patents. We adopted the patent citation style of Recent Patents on Inflammation & Allergy Drug Discovery.

2) Line 70 - explain the term "estimated public benefit" in greater detail.

Answer: The sentence was amended: “The estimated public benefits in terms of a higher reduction of the disease burden of campylobacteriosis and its sequelae are greater if efforts are made towards controlling C. jejuni during the primary production stage.”

3) Line 125 - The phrase "environmentally friendly products" needs to be described in full.

Answer: We cancelled the word environmentally friendly in this sentence, because we did not mean environmentally friendly in the meaning of avoiding pollution. What we want to express that an antibiotic-free method does not put evolutionary pressure on bacteria resulting in multiresistant strains in the environment/waste water.

Thus, the sentence was rephrased to: “Phage-based applications can be used as a save alternative to antibiotics in live animals, in situ on processed meat and other food, as well as for decontamination of surfaces and equipment in food processing to ensure that no muliresistant bacterial strains selected as a result of antibiotic-based evolutionary pressure are released into the wastewater and thus into the environment.”

4) Table 2 - in the Wagenaar et al section, you use the phrase "went up again". This needs to be changed to rephrased.

“went back up again in 5 days” was replaced by “increased again within 5 days”

5) Table 2 - you alternate between log and log 10 in different sections. This needs to be clarified.

Answer: We have taken the information directly, also in the way it is written, from the original publications. Since the decadic logarithm log10 is usually simply given as log even without specifying the base, we have mixed the notations according to the original publications. However, our publication always refers to the decadic logarithm. In order to use a mathematically precise, uniform notation, we have now converted all corresponding specifications with base 10 into log10.

6) Line 285 - rephrase "bounce back".

“bounce back” was replaced by “rise again”

Reviewer 2 Report

General Comments

The manuscript reviews the current research on use of bacteriophages to control Campylobacter jejuni. The manuscript is pertinent and provides generalized overview of the field.

However, there are issues that needs to be addressed. Certain sections within the manuscript require extensive modifications. The manuscript may also benefit from English language editing.

Section 4.1; needs to be edited for uniformity. In its current form it appears bullet points are subsection of the first study presented. Section 4.1; This section needs a thorough discussion on the nature and outcome of the studies presented, their shortcomings and a projection of further research. Currently this section presents only paraphrases of the study summaries. Section 4.2; Similar to section 4.1, this section also needs a thorough discussion of the studies presented, their outcome and projection of future research. Section 5.2; This section needs to be rearranged along the three resistance types and not by different studies. The three resistance mechanisms need to be clearly defined and studies pertaining each resistance mechanism should be discussed within same subsection. A section on “Handling the resistance” should be included as the main heading of the Section 5 suggests.

Specific Comments

L 64-65; the sentence needs to be revised to clarify the meaning. L65-68; Again the sentence needs to be revised. Do authors suggest the various intervention measures are ineffective? L4-75; The statement “Campylobacter phages …………. under check.” Is too definitive; Has this been established for a fact that application of bacteriophages keeps C. jejuni under control? L78; “….. therapeutic phages [27,28].” Please use correct citations for the text. The two citations presented here do not support therapeutic usage of phages. Sacher, J.C.; Yee, E.; Szymanski, C.M.; Miller, W.G. Complete Genome Sequences of Three Campylobacter jejuni Phage-propagating Strains. Genome Announc 2018, 6, 1–2. Frost, J.A.; Kramer, J.M.; Gillanders, S.A. Phage typing of Campylobacter jejuni and Campylobacter coli and its use as an adjunct to serotyping. Epidemiol Infect 1999, 123, 47–55. L 83 and elsewhere; The organism names have to be in italics. L93; Is this classification stands only for ‘Lytic bacteriophages’ or all the Campylobacter L104; remove “that”. L115-130; This paragraph appears to be lacking the most likely and direct reasoning – ability of phages to target resistant bacterial pathogens in similar fashion as non-resistant pathogen. Please discuss the relation of phages and antimicrobial resistant pathogens. L134; Please discuss what is meant here by “Due to this intrinsic safety,….”. The presence of phages does not imply that phages are safe. L137-139; Delete unnecessary information beginning “which made ………… approval.” L142; revise statement “against Shigella-states specifically …” to “against Shigella, specifically states ….” Correct spelling of phages. L144-152; A more thorough discussion on lysogenic phages are required. The use of sequencing and bioinformatics can be placed in a new paragraph and discussed in detail. L155-156; The sentence “According to …………. hospitalizations.” is vague and needs more specific information. L165; The statement “Progress … prepared….” is incorrect, the therapeutic success is not present in the US, hence, was not the background for regulatory approvals of phage-biocontrol applications. L194-195; Please add citations. In authors’ opinion, what might be the explanation for such varied levels of reductions of jejuni? L241; Please be consistent in the presentation of "Conclusion". Section 4.1 does not have a conclusion. Section 5.1; The problem/s related to isolation and characterization needs to be defined and clearly stated. The authors are advised to provide a paragraph stating the problems encountered in isolation and characterization of Campylobacter L285; What is the pretext for this statement “The reason …………. “. The sentence appears to be without any context.

Author Response

Reviewer II

Section 4.1; needs to be edited for uniformity. In its current form it appears bullet points are subsection of the first study presented. Section 4.1; This section needs a thorough discussion on the nature and outcome of the studies presented, their shortcomings and a projection of further research. Currently this section presents only paraphrases of the study summaries.

Answer: Section 4.1 was significantly revised and expanded according to the reviewer's suggestions.

Section 4.2; Similar to section 4.1, this section also needs a thorough discussion of the studies presented, their outcome and projection of future research.

Answer: Section 4.2 was significantly revised and expanded according to the reviewer's suggestions.

Section 5.2; This section needs to be rearranged along the three resistance types and not by different studies. The three resistance mechanisms need to be clearly defined and studies pertaining each resistance mechanism should be discussed within same subsection. A section on “Handling the resistance” should be included as the main heading of the Section 5 suggests.

 Answer: Section 5.2 was significantly revised and expanded according to the reviewer's suggestions.

Specific Comments

L 64-65; the sentence needs to be revised to clarify the meaning.

Answer: The sentence was changed to: The colonization of broiler chicks by C. jejuni takes place within the first three weeks after hatching, and since Campylobacter colonization of chicks is not associated with signs of disease, the horizontal spread usually remains unnoticed.

L65-68; Again the sentence needs to be revised. Do authors suggest the various intervention measures are ineffective?

Answer: The two sentences were changed to: Combination of strict biosecurity, good medical practice (GMP) and hazard analysis and critical control points (HACCP) attempts can alleviate contamination levels but do not lead to a complete elimination of the bacteria. It is difficult to quantify the effectiveness of the individual measures because they are dependent on very many interrelated local factors.

L74-75; The statement “Campylobacter phages …………. under check.” Is too definitive; Has this been established for a fact that application of bacteriophages keeps C. jejuni under control?

Answer: The statement was toned down to: “Campylobacter phages are natural "predators" that could potentially be able to bring C. jejuni under control.”

L78; “….. therapeutic phages [27,28].” Please use correct citations for the text. The two citations presented here do not support therapeutic usage of phages.

Sacher, J.C.; Yee, E.; Szymanski, C.M.; Miller, W.G. Complete Genome Sequences of Three Campylobacter jejuni Phage-propagating Strains. Genome Announc 2018, 6, 1–2.

Frost, J.A.; Kramer, J.M.; Gillanders, S.A. Phage typing of Campylobacter jejuni and Campylobacter coli and its use as an adjunct to serotyping. Epidemiol Infect 1999, 123, 47–55.

Answer: The right reference is: Zampara, A.; Sørensen, M.C.H.; Elsser-Gravesen, A.; Brøndsted, L. Significance of phage-host interactions for biocontrol of Campylobacter jejuni in food. Food Control 2017, 73, 1169–1175. This was corrected:

L 83 and elsewhere; The organism names have to be in italics.

We corrected the italic writing of bacterial species names in this section of the edited manuscript. However, the bacterial species names in the original manuscript uploaded by us were already in italics.

L93; Is this classification stands only for ‘Lytic bacteriophages’ or all the Campylobacter

This classification stands for Lytic Campylobacter phages – “Lytic” was added.

L104; remove “that”.

Answer: The sentence was changed to: “Sørensen and colleagues showed, with some exceptions, the connection between receptor dependency and grouping of C. jejuni phages.”

L115-130; This paragraph appears to be lacking the most likely and direct reasoning – ability of phages to target resistant bacterial pathogens in similar fashion as non-resistant pathogen. Please discuss the relation of phages and antimicrobial resistant pathogens.

Answer: see also Reviewer I point 3: The missing aspect of the independence of phage action from the susceptibility of host bacteria to antibiotics was included in the passage.

L134; Please discuss what is meant here by “Due to this intrinsic safety,….”. The presence of phages does not imply that phages are safe.

Answer: As the reviewer notes, the statement lacks a logical coherence. The classification as generally regarded as safe (GRAS) is of course not based on “Intinsic Safety”, but on the evaluation of experts qualified by scientific training and experience.

L137-139; Delete unnecessary information beginning “which made ………… approval.”

Answer: “, which made the substances and chemicals historically used for flavoring or curing foods prior to 1958 and the substances that were generally recognized, among qualified experts, as safe under the conditions of their intended use, exempt from premarket approval” was deleted.

L142; revise statement “against Shigella-states specifically …” to “against Shigella, specifically states ….” Correct spelling of phages.

Answer: The statement has been changed as suggested and the “h” of phages was introduced in L144.

L144-152; A more thorough discussion on lysogenic phages are required. The use of sequencing and bioinformatics can be placed in a new paragraph and discussed in detail.

Answer: A new section “Determining Safety of Phages in the Era of “Omics”” was added.

L155-156; The sentence “According to …………. hospitalizations.” is vague and needs more specific information.

Answer: The statement was replaced by: “According to the US CDC 841 foodborne disease outbreaks were reported by 50 states, Washington, D.C., and Puerto Rico, resulting in 14,481 illnesses, 827 hospitalizations, 20 deaths, and 14 food recalls in 2017 [CDC 2017, https://www.cdc.gov/fdoss/pdf/2017_FoodBorneOutbreaks_508.pdf]”

L165; The statement “Progress … prepared….” is incorrect, the therapeutic success is not present in the US, hence, was not the background for regulatory approvals of phage-biocontrol applications.

Answer: The stamen was corrected, that means replaced by “The general recognition of phage applications as GRAS prepared also the ground for regulatory…”.

L194-195; Please add citations. In authors’ opinion, what might be the explanation for such varied levels of reductions of C. jejuni?

Answer: The references were added. Reasons for varied levels of reductions have been explained in section 5.2.

-efficacy of different groups of phages

-host strain diversity

-Resistance

-use of phages from cell banks as opposed to isolated strains

L241; Please be consistent in the presentation of "Conclusion". Section 4.1 does not have a conclusion.

We deleted the Conclusions at the end of this section.

Section 5.1; The problem/s related to isolation and characterization needs to be defined and clearly stated. The authors are advised to provide a paragraph stating the problems encountered in isolation and characterization of Campylobacter

Answer: In contrast to the reviewer we do not see real “problems” in isolation and characterization of Campylobacter isolates. There are well establishes selective media e.g. CAM-Agar, and typing methods like MALDI-TOF as well as MLST. In line with the other sections, we have removed the bullet points and made slight modifications.

L285; What is the pretext for this statement “The reason …………. “. The sentence appears to be without any context.

Answer: The Pretext of this statement is now given in the new subsection 5.2.2

Round 2

Reviewer 2 Report

No comments